# GABA and Combined GABA with GAD65-Alum Treatment Alters Th1 Cytokine Responses of PBMCs from Children with Recent-Onset Type 1 Diabetes

**DOI:** 10.3390/biomedicines11071948

**Published:** 2023-07-10

**Authors:** Katie E. Heath, Joseph M. Feduska, Jared P. Taylor, Julie A. Houp, Davide Botta, Frances E. Lund, Gail J. Mick, Gerald McGwin, Kenneth L. McCormick, Hubert M. Tse

**Affiliations:** 1Department of Microbiology, Comprehensive Diabetes Center, University of Alabama at Birmingham, Birmingham, AL 35294, USAjfeduska@gmail.com (J.M.F.); jptaylor@uab.edu (J.P.T.); dbotta@uab.edu (D.B.); flund@uab.edu (F.E.L.); 2Department of Surgery, Heersink School of Medicine, University of Alabama at Birmingham, Birmingham, AL 35294, USA; jhoup@uab.edu; 3Department of Pediatrics, Division of Pediatric Endocrinology, University of Alabama at Birmingham, Birmingham, AL 35294, USA; gjmick@uabmc.edu (G.J.M.); klmccormick@uabmc.edu (K.L.M.); 4Department of Epidemiology, School of Public Health, University of Alabama at Birmingham, Birmingham, AL 35294, USA; gmcgwin@uabmc.edu; 5Department of Microbiology, Molecular Genetics, and Immunology, University of Kansas Medical Center, Mail Stop 3029, 1012 Wahl Hall West, 3901 Rainbow Boulevard, Kansas City, KS 66160, USA

**Keywords:** type 1 diabetes, GABA, autoimmunity, GAD65, cytokine

## Abstract

Type 1 diabetes (T1D) is an autoimmune disease culminating in the destruction of insulin-producing pancreatic cells. There is a need for the development of novel antigen-specific strategies to delay cell destruction, including combinatorial strategies that do not elicit systemic immunosuppression. Gamma-aminobutyric acid (GABA) is expressed by immune cells, β-cells, and gut bacteria and is immunomodulatory. Glutamic-acid decarboxylase 65 (GAD65), which catalyzes GABA from glutamate, is a T1D autoantigen. To test the efficacy of combinatorial GABA treatment with or without GAD65-immunization to dampen autoimmune responses, we enrolled recent-onset children with T1D in a one-year clinical trial (ClinicalTrials.gov NCT02002130) and examined T cell responses. We isolated peripheral blood mononuclear cells and evaluated cytokine responses following polyclonal activation and GAD65 rechallenge. Both GABA alone and GABA/GAD65-alum treatment inhibited Th1 cytokine responses over the 12-month study with both polyclonal and GAD65 restimulation. We also investigated whether patients with HLA-DR3-DQ2 and HLA-DR4-DQ8, the two highest-risk human leukocyte antigen (HLA) haplotypes in T1D, exhibited differences in response to GABA alone and GABA/GAD65-alum. HLA-DR4-DQ8 patients possessed a Th1-skewed response compared to HLA-DR3-DQ2 patients. We show that GABA and GABA/GAD65-alum present an attractive immunomodulatory treatment for children with T1D and that HLA haplotypes should be considered.

## 1. Introduction

Type 1 diabetes (T1D) is an autoimmune disease resulting in the destruction of insulin-producing β-cells in the pancreas. Patients with T1D are unable to properly regulate blood glucose homeostasis without exogenous insulin treatment and are susceptible to long-term complications such as blindness, cardiovascular disease, cerebral disorders, and neuropathies [1]. While β-cell replacement therapies and immunoprotection strategies have been refined in the past decade, there is still a pressing need to identify novel approaches for delaying or reversing autoreactive T cell responses. Current immunotherapies efficacious in protecting β-cell function include anti-CD3 monoclonal antibody therapy (Teplizumab) [2], antithymocyte globulin (ATG) treatment with granulocyte colony-stimulating factor (G-CSF) [3], Rituximab [4], and adoptive transfer with autologous regulatory T cells (Treg) [5,6,7]. However, systemic immunosuppression place children at risk of microbial infections, and large subgroups of patients are refractory to these treatments, highlighting the need for more targeted approaches.

Since Th1 cytokine responses and CD8 effector T cell (Teff) infiltration of islets drive T1D progression, targeting antigen-specific autoreactive T cells is an optimal approach to preserve remaining β-cell function and delay recurring autoimmunity. Several putative autoantigens have been implicated in T1D development, including insulin [8], islet amyloid polypeptide (IAPP) [9], chromogranin A [10], zinc transporter 8 [11], and glutamic acid decarboxylase 65 (GAD65) [12]. GAD65-specific autoreactive T cells promote β-cell destruction [13], and various clinical trials have targeted GAD65-specific T cells by immunizing with alum-formulated-GAD (GAD65-alum) to delay the autoimmune destruction of β-cells. The efficacy of GAD65-alum in preserving endogenous insulin production through measuring c-peptide has been highly debated. A 2008 clinical trial studying recent-onset patients with T1D showed that GAD65-alum treatment preserved residual c-peptide [14], but two ensuing trials showed no effect of GAD65-alum on c-peptide [15,16]. However, a subsequent meta-analysis of these trials concluded with a high probability that GAD65-alum administration yielded a positive effect on c-peptide levels [17,18], and an updated meta-analysis and recent phase 2b trial suggest efficacy in 40–50% of patients carrying HLA-DR3-DQ2, especially when administered intralymphatically [19,20]. The capacity of GAD65-alum to regulate immune responses has been corroborated in many studies, which showed a Th2/Treg-skewed immune cell profile [21,22,23].

In addition to GAD65-alum, the use of gamma-aminobutyric acid (GABA) can also hinder proinflammatory immune responses. Immune cells from both the innate (myeloid) and adaptive (lymphoid) lineages, such as macrophages and CD8 T cells [24], respectively, express GABA_A_ receptors (GABAARs) [25]. Through activation of the phosphatidylinositol 3-kinase (PI3K)/Akt axis, GABA treatment can dampen inflammatory Teff cells and enhance Treg populations [26]. GABA is also synthesized by pancreatic β-cells, resulting in both decreased glucagon secretion by α-cells and prolonged β-cell survival [26,27,28]. Additionally, treatment of human peripheral blood mononuclear cells (PBMCs) in vitro with GABA can inhibit the release of Th1- and Th2-specific proinflammatory cytokines, particularly in PBMCs from patients with T1D compared to healthy controls [29].

Based on the effects of GABA on immune cells and pancreatic β-cells, we hypothesize that GABA and GAD65-alum will reduce immune inflammatory responses in children with T1D. Our results show that GABA with GAD65-alum treatment dampens proinflammatory Th1 cytokine responses compared to patients receiving placebo control. Furthermore, we characterized the study participants based on their expression of the T1D high-risk alleles HLA-DR3-DQ2.5 and HLA-DR4-DQ8.1. Patients possessing both alleles or homozygous for either are considered high risk [30]. We demonstrated that patients possessing at least one HLA-DR4-DQ8.1 allele are predisposed to an elevated inflammatory phenotype at diagnosis and also respond well to GABA treatment. Taken together, these results show the advantages of using GAD65-specific therapy in children with T1D and the importance of HLA haplotypes in determining patient outcomes.

## 2. Materials and Methods

### 2.1. Study Design

This study was an analysis of immune cell function using a cohort of children with recent-onset T1D. Participants were enrolled in the primary study (ClinicalTrials.gov Identifier: NCT02002130; IRB Protocol #: F130807009) at the University of Alabama at Birmingham [31]. Patient characteristics are described in Table 1. All patients tested positive for anti-GAD65 autoantibodies. GAD65-alum was administered as a 20 μg primary injection at baseline, with a 20 μg booster 1 month later. The sample size for the proposed study is 95 children; 25 in the active GABA/GAD-alum treatment group, 35 in the GABA/Placebo Gad-alum group, and 35 in the placebo group. For the primary comparison of the 12-month post-baseline C-peptide measurements between these groups, assuming an α of 0.05 and a mean (SD) C-peptide AUC of 1.0 (0.4), this sample size yields a ~97% power to detect a 50% difference.

### 2.2. Participants and Eligibility Criteria

Participants were screened at the time of diagnosis with T1D, as defined by ADA criteria. All patients were enrolled from the clinics and in-patient wards at Children’s of Alabama (CoA), a tertiary care university-associated referral center. The majority of patients were residents of the state of Alabama. There were 11 out-of-state participants (AZ, GA, MS, MO, NC, ND, TX, and VA). The first participant was enrolled 2 March 2015, and the last study visit was 24 June 2019. Inclusion criteria: children 4–18 years of age, positivity for autoantibody GAD65, and enrollment within 5 weeks of diagnosis. If the participant was female and not abstinent, two forms of contraception were required. Exclusion criteria: pregnancy, systemic or inhaled steroid use, neurologic/seizure disorders, adjunct oral therapies that might affect glucose or GABA metabolism. Six randomized patients were excluded from the analysis because all c-peptide values, fasting, and MMTT stimulated were <0.6 ng/mL at the initial baseline study visit. Participants received a $60 gift card as compensation for every blood draw.

### 2.3. PBMC Collection and Stimulation

Human PBMCs were purified from blood using Lymphoprep (STEMCELL Technologies, Vancouver, BC, Canada) in SepMate PBMC isolation tubes (STEMCELL Technologies). Samples were collected at baseline, 5 months, and 12 months after beginning treatment. Purified PBMCs were frozen in liquid nitrogen with FBS (Atlas Biologicals, Ft. Collins, CO, USA) with 10% DMSO (Sigma, St. Louis, MO, USA) until all time point samples from each patient were collected. PBMCs were then thawed and stimulated (10^5^) with either Dynabeads (Gibco, Carlsbane, CA, USA) or rhGAD65 (5 μg/mL, Diamyd, Stockholm, Switzerland) in the presence of IL-2 (30 U/mL) in DMEM media (Life Technologies, Carlsbane, CA, USA) containing FBS (Atlas), human AB serum (Fisher, Hampton, NH, USA), and penicillin/streptomycin (Gibco) for 3 days (Dynabeads) or 7 days (rhGAD65). Cell culture supernatants were collected and cells were stored in RNAlater (Thermo Fisher, Waltham, MA, USA).

### 2.4. Quantitative RT-PCR

mRNA was isolated using RNeasy Mini Kit (Qiagen, Hilden, Germany) and converted to cDNA using a high-capacity cDNA reverse transcription kit (Thermo Fisher). cDNA was quantitated using TaqMan gene expression assays (Applied Biosystems, Waltham, MA, USA, Table 2) using a Roche LightCycler 480 II. *GAPDH* was used for normalization, and the 2^−∆∆Ct^ method for calculating relative gene expression

### 2.5. Milliplex Cytokine and Chemokine Analysis

Cell culture supernatants were analyzed with a Milliplex MAP human cytokine/chemokine bead panel (Millipore-Sigma, Burlington, MA, USA, Table 3). Results are presented as mean ± SD.

### 2.6. Patient HLA Genotyping

Genomic DNA was isolated from frozen PBMC samples with the DNeasy Blood and Tissue Kit (Qiagen) according to the manufacturer’s instructions. HLA typing was performed by the Histocompatibility and Immunogenetics Laboratory at the University of Alabama at Birmingham. Low-resolution molecular typing of HLA class II loci (DRB1 and DQB1) was performed using Luminex-based LabType SSO (One Lambda, West Hills, CA, USA). Patients were assigned to either the HLA-DR3-DQ2 group or the HLA-DR4/other group. Of the 51 patient samples, 30 possessed the HLA-DR3-DQ2 allele (with two patients being homozygous). Of the remaining 21 patients, 14 had at least 1 copy of HLA-DR4-DQ8, and the remaining 7 had neither risk allele. To recapitulate the previous studies demonstrating HLA-DR3-DQ2 responded to GAD65-alum to increase c-peptide and insulin levels [19,20], we grouped the patients as either HLA-DR3-DQ2 or HLA-DR4-DQ8/Other (Table 4).

### 2.7. Statistical Analysis

All data are shown as mean ± standard deviation (SD). Kruskal–Wallis tests were used to compare the measurements between the groups at each time point. Signed rank tests were used to compare measurements between time points within each group.

## 3. Results

### 3.1. GABA Treatment Decreases Th1 Cytokine and Chemokine mRNA Accumulation in Polyclonal-Stimulated PBMCs

The goal of this study was to determine whether oral GABA treatment alone or GABA supplemented with GAD65 immunizations could improve both clinical [31,32] and immunological outcomes for children with T1D. Since GAD65 is a T1D autoantigen [12], we hypothesized that GABA and GAD65-alum interventions would blunt proinflammatory effector T cell (Teff) responses in children with T1D. We separated our patient cohort into three experimental groups [32]. Patients received oral GABA twice daily, oral GABA plus injections of 20 mg rhGAD65 (GAD65-alum) at baseline and 1-month, or placebo. PBMCs were collected at baseline, 5 months, and 12 months and evaluated for changes in proinflammatory and anti-inflammatory cytokine and chemokine responses. We first performed mRNA analysis on PBMC samples following polyclonal stimulation for 72 h with anti-CD3/CD28 Dynabeads. We observed a significant (*p* < 0.05) decrease in *IFNG* expression in the GABA treatment group at 5 months compared to the placebo group and the GABA with GAD65-alum group at 12 months (*p* < 0.05) compared to placebo (Figure 1A). The GABA with GAD65-alum group showed a trending decrease in *IFNG* expression compared to the GABA group (*p* = 0.0553). Compared to placebo, the GABA group did not show significant differences in *IL21* (Figure 1B) or *CXCL10* mRNA accumulation (Figure 1C). However, the GABA with GAD65-alum group showed significant (*p* < 0.05) increases in expression of both *IL21* and *CXCL10* at 5 months compared to GABA only (Figure 1B,C). Analysis of additional proinflammatory genes including *TNF*, *BCL6*, and *CCL5* showed no significant changes in expression across all groups and time points (Appendix A). The master transcriptional regulator of Treg differentiation, *FOXP3*, showed increased expression at 12 months in the GABA group compared to the placebo (Figure 1D), but not in the GABA with GAD65-alum group. The anti-inflammatory cytokine *IL10* (Figure 1E) and inhibitory immune receptor *CTLA4* (Appendix A) showed no significant changes in mRNA expression. Our results demonstrate that treatment with GABA, particularly with the addition of GAD65-alum, decreased Th1 cytokine responses (*IFNG*) of polyclonal-stimulated PBMCs in comparison to the placebo group. Additionally, GABA treatment did not affect anti-inflammatory cytokine synthesis (*IL10*) but enhanced *FOXP3* mRNA accumulation.

### 3.2. Treatment with GABA and GAD65-Alum Skews PBMC Response with Antigen-Specific Stimulation Away from a Th1 Profile

To further elucidate the effect of GABA with GAD65-alum on antigen-specific responses, we treated PBMCs from each group with rhGAD65 in an antigen-recall assay. PBMCs were isolated from patients at the baseline, 5-month, and 12-month time points, and gene expression was analyzed following 7 days of in vitro stimulation with rhGAD65. We observed a significant decrease in *IFNG* mRNA accumulation in the GABA with GAD65-alum group at 12 months compared to both placebo and the GABA alone treatment group (Figure 2A). The GABA with GAD65-alum showed a decrease in *IFNG* mRNA accumulation compared with the GABA treatment group at the 5-month time point, though this difference was just above the threshold for significance (*p* = 0.0569). The proinflammatory genes *IL21* (Figure 2B), *CXCL10* (Figure 2C), *TNF* (Figure 2D), and *CCL5* (Figure 2E) demonstrated no significant changes in expression, except for significantly lower *TNF* expression in the GABA group at baseline compared to placebo. With rhGAD65 stimulation, no significant changes were detected in *IL10* (Figure 2F), *FOXP3* (Figure 2G), *CTLA4* (Figure 2H), or *BCL6* (Appendix A) mRNA accumulation across all groups or at the various time points. These results corroborated our polyclonal-stimulated PBMC studies, as patients with T1D treated with GABA and GAD65-alum demonstrated a dampened Th1 (*IFNG*) cytokine response following rhGAD65 stimulation.

### 3.3. Longitudinal Analysis Shows That GABA and GABA with GAD65-Alum Treatment Halts Progression of Inflammatory Phenotype

We next confirmed changes in PBMC mRNA expression by analyzing cytokine and chemokine secretion from the cell culture supernatants of the rhGAD65 antigen-recall assay for 7 days. We did not initially observe significant differences in key Th1 effector cytokine and chemokine (IFN-γ, TNF, and CXCL10) synthesis between placebo, GABA treatment, and GABA plus GAD65-alum treatment groups. However, when we compared each treatment group at 5 months and 12 months to their respective baseline time point, we found evidence supporting a blunted Th1 cytokine and chemokine response following GABA with GAD65-alum treatment. The placebo group showed a significant increase (*p* < 0.05) in IFN-γ synthesis from the 5-month to 12-month interval and from baseline to 12 months (*p* < 0.01, Figure 3A). However, the GABA and the GABA with GAD65-alum groups showed no significant change in secreted IFN-γ from baseline to 5 months and to 12 months (Figure 3A), suggesting that proinflammatory Th1 effector responses did not increase over time following treatment. The placebo group showed an increase (*p* < 0.05) in TNF from baseline to 12 months, while there was no change in the GABA or GABA with GAD65-alum groups (Figure 3B). Patients receiving a placebo displayed a significant increase in CXCL10 from 5 months to 12 months (*p* < 0.05) and an overall increase (*p* < 0.01) from baseline to 12 months (Figure 3C). Conversely, patients receiving GABA treatment showed a significant decrease from baseline to 5 months (*p* < 0.05) but an increase (*p* < 0.0001) from 5 months to 12 months (Figure 3C). There were no significant CXCL10 differences observed in the GABA with GAD65-alum group (Figure 3C). We also examined IL-12p70 and IL-12p40 synthesis, which are necessary for IFN-γ production and Th1 lineage commitment. The placebo and GABA groups did not show differences in secreted IL-12p40 over time from baseline to 12 months (Figure 3D). However, the GABA with GAD65-alum group showed a significant (*p* < 0.05) reduction in IL-12p40 between 5 months and 12 months (Figure 3D). Similarly, we observed no differences in IL-12p70 in the placebo or GABA groups, but there was a decrease (*p* = 0.0566) in the GABA with GAD65-alum group at the 12-month time point (Figure 3E).

Differences were observed in other cytokine responses. Placebo patients showed a significant (*p* < 0.05) increase in IL-6 from baseline to 12 months (Figure 3F). GABA patients also showed an increase from baseline to 5 months (*p* < 0.05) but then showed a decrease by 12 months, leaving the modest increase from baseline to 12 months not significant (Figure 3F). Interestingly, patients receiving GABA with GAD65-alum started with higher baseline IL-6 that decreased over the 5-month to 12-month interval (Figure 3F). The placebo group showed a significant increase (*p* < 0.05) in IL-8 (CXCL8) from 5 months to 12 months, with an overall increase from baseline to 12 months (Figure 3G). Patients receiving GABA alone or GABA plus GAD65-alum did not display any significant changes in IL-8 synthesis from baseline to 5- or 12-month time points.

Since IL-2 is necessary for the proliferation and expansion of effector and memory Th1 CD4 T cells [33], we examined if IL-2 levels varied between treatment groups. We observed a significant decrease in IL-2 production by PBMCs in the placebo group from baseline to 5 months (*p* < 0.05) and 12 months (*p* < 0.05) (Figure 3H). In the GABA group, we also observed a significant decrease from baseline to 5 months (*p* < 0.05), but by 12 months, IL-2 levels significantly increased (*p* < 0.05) back to baseline. PBMCs from patients receiving GABA plus GAD65-alum showed no significant differences between time points (Figure 3H). Taken together, our qPCR and Milliplex results indicate that GABA or GABA with GAD65-alum treatment can inhibit the synthesis of some proinflammatory cytokines and chemokines from polyclonal and rhGAD65-stimulated PBMCs.

### 3.4. Genotyping Reveals Differences in Basal Inflammatory Predisposition and Response to GABA Treatment Based on HLA

As genetics contribute to disease susceptibility, including HLA class II [34], genetics may also play a role in determining treatment efficacy. Patients with the HLA-DR3-DQ2 haplotype respond better to GAD65-alum treatment, as demonstrated by increased c-peptide retention [19,20]. To investigate whether the HLA-DR3-DQ2 haplotype was a determining factor in immune responses following GABA or GABA with GAD65-alum treatment in children with T1D, we performed HLA-typing on our patient samples and divided our data into their respective HLA haplotypes. HLA-typing showed that 30 patients possessed the high-risk HLA-DR3-DQ2 genotype (abbreviated HLA-DR3), while the remaining 21 patients were heterozygous for HLA-DR4-DQ8 (14 patients) or other allele combinations (HLA-DR4/Other) (Table 3). While the data from polyclonal stimulation did not yield any significant differences based on HLA haplotypes, GAD65-stimulated PBMCs illuminated several significant differences based on HLA haplotype. We observed that HLA-DR3 patients receiving GABA with GAD65-alum treatment for 12 months not only had significantly lower *IFNG* compared to placebo (*p* < 0.001) and GABA alone (*p* < 0.01) but were also significantly lower (*p* < 0.05) than their HLA-DR4/Other counterparts who also received the same treatment (Figure 4A). We also observed in HLA-DR3 patients that *CXCL10* was significantly decreased (*p* < 0.05) in the GABA with GAD65-alum group compared to the HLA-DR3 placebo and the HLA-DR4/Other GABA group at 12 months (Figure 4B). Interestingly, at baseline, GABA and GABA with GAD65-alum patients with HLA-DR3 were significantly lower (*p* < 0.05) in *TNF* compared to their counterparts in the placebo group (Appendix A). These differences were lost at 5 months and 12 months post-treatment. No changes in *IL21*, *FOXP3*, *IL10*, *BCL6*, *CCL5,* and *CTLA4* mRNA accumulation were noted at the 5-month or 12-month time points (Appendix A).

We then corroborated our mRNA findings by stratifying our Milliplex results along the HLA haplotype. We found HLA-DR4/Other patients receiving GABA treatment secreted significantly more GM-CSF (*p* < 0.05) and IL-1α (*p* < 0.01) than HLA-DR3 at baseline, but this effect was diminished at 5 months and 12 months (Appendix A). We next examined Th1-specific responses, which are characteristic of T1D pathogenesis. Patients in the placebo group did not show statistically significant differences in IFN-γ; however, patients with the HLA-DR4/Other genotype trended higher than HLA-DR3 patients at all time points (Figure 4C). HLA-DR4/Other patients in the GABA group started with significantly (*p* < 0.05) higher IFN-γ at baseline than HLA-DR3 counterparts (Figure 4C), but this difference diminished over the course of treatment. HLA-DR4 patients receiving GABA with GAD65-alum displayed increased IFN-γ (*p* < 0.05) compared to HLA-DR3 by 12 months (Figure 4C). CXCL10 was significantly elevated (*p* < 0.05) in the GABA group from HLA-DR4/other patients compared to HLA-DR3 at baseline, but no differences were observed at the 5-month and 12-month time points (Figure 4D). TNF was also significantly elevated (*p* < 0.01) in PBMCs from HLA-DR4/Other patients compared to HLA-DR3 at baseline but appeared to be increasing again by 12 months (*p* = 0.0559) (Figure 4E). Furthermore, after stratifying by HLA, PBMCs from HLA-DR3 patients in the GABA treatment group secreted less TNF (*p* < 0.05) than HLA-DR3 patients in the placebo group at 12 months (Figure 4E). IL-12p70 was also found to be elevated (*p* < 0.05) in HLA-DR4/Other patients at baseline in the GABA treatment group but did not increase over time (Figure 4F). No significant differences were observed in placebo or GABA with GAD65-alum groups. Together, these data suggest that patients with HLA-DR4/Other are predisposed to an elevated Th1 phenotype compared to HLA-DR3 patients but respond well to GABA treatment.

## 4. Discussion

Antigen-specific immunotherapy is an ideal treatment option for autoimmune diabetes by inhibiting autoreactive T cell responses while minimizing off-target side effects. In this study, we examined the efficacy of orally administered GABA with and without GAD65-alum antigen immunization to suppress proinflammatory autoimmune responses in children with recently diagnosed T1D. We evaluated the immune response of PBMCs from 97 patients randomized in the UAB GABA/GAD Clinical Trial following polyclonal and antigen-specific stimulation with rhGAD65.

The key findings from our PBMC analysis were the inhibition of *IFNG* mRNA accumulation following polyclonal and GAD65 antigen-specific stimulation by GABA with and without GAD65-alum treatment, as well as a delay in the longitudinal increase in IFN-γ synthesis following GABA with GAD65-alum treatment. This is significant since IFN-γ can activate macrophages and CD8 T cells to facilitate β-cell destruction in T1D [35,36]. IFN-γ also stimulates the release of IFN-inducible chemokines such as CXCL10 by β-cells [37]. Notably, we observed a transient inhibition of CXCL10 with GABA treatment and no increase in GABA with GAD65-alum treatment. The decrease in CXCL10 synthesis may have profound effects on T cell migration and infiltration in islets and further delay β-cell destruction in patients with T1D [38,39]. Despite being a hallmark cytokine for Th1 responses, IFN-γ inhibition should not be viewed as a complete end-all, for under certain circumstances, IFN-γ can have inhibitory effects on CD8 T cell responses in T1D [40]. Our results are in concordance with another outcome measure from this study in that the escalating Th1 signature in the placebo group coincided with an increasing ratio of proinsulin to c-peptide [41].

Accumulating evidence has demonstrated that Th17 or T follicular helper (Tfh) cells may also contribute to T1D immunopathogenesis [42,43,44]. IL-6 is a pleiotropic cytokine responsible for mediating the maturation of Th17 effector cells and Tfh cells and inhibiting Treg development [45]. Similar to our observation with IFN-γ, the placebo group displayed increased IL-6 production over 12 months, but this was halted in both GABA treatment groups. Future studies will examine whether Tfh or Th17 differentiation is affected by GABA and GAD65-alum treatment by purifying and immunophenotyping these T cell subsets from the peripheral blood.

IL-2 is required for the expansion and maturation of naïve T cells to Teff and for Treg maintenance [33]. PMBCs from the placebo group showed an overall significant decrease in IL-2 between baseline and 12 months and a significant increase from 5 months to 12 months. Similarly, GABA treatment showed a significant decrease from baseline to 5 months, then increased from 5 months to 12 months. Decreased IL-2 at 5 months in the placebo and GABA groups may partially corroborate the increase in IFN-γ synthesis observed in these patients due to an increase in the differentiation of Teff cells. In contrast, the unchanged IL-2 levels observed in the GABA with GAD65-alum group may provide evidence that Treg differentiation is increased and/or Teff cells are not fully differentiated. One limitation of our study is that our immunophenotyping of PBMCs may not recapitulate the immune response present in the islet microenvironment in children with T1D. Further studies are warranted in animal models of T1D treated with GABA and GAD65-alum to corroborate our human immunophenotyping results and also examine if peripheral immune responses resemble the islet microenvironment.

Successful cures for T1D must restore functional β-cells and inhibit the recurrence of autoimmunity. Our results corroborate other studies investigating the efficacy of GABA immunosuppression in murine models of T1D. Soltani et al. showed that GABA inhibited in vitro IL-12 production by macrophages, IFN-γ by CD4 T cells, and circulating levels of IL-12, IFN-γ, TNF, and IL-1β in multiple low-dose streptozotocin-induced diabetes model [26]. Apart from suppressing autoimmune responses, two separate studies showed that GABA could regenerate β-cells [46,47]. Ben-Othman et al. administered GABA weekly for a period of 6 months to streptozotocin-treated mice and successfully converted α-cells to β-like cells from ductal precursors. This was partially due to GABA_A_ receptor activation and decreased *Arx* expression to control α and β-cell fate [46]. Li et al. similarly showed that the drug artemether released gephyrin from the GABA_A_ receptor complex, which also leads to ARX suppression and β-like cell reprogramming from α-cells [47]. These studies hold exciting promise that GABA can successfully rescue β-cell mass in T1D; however, neither study addressed the underlying autoimmune response.

There are a few reports combining GABA-mediated β-cell regeneration and immunosuppression in NOD mice. Tian et al. combined GABA with proinsulin-alum to restore normoglycemia and inhibit T cell responses in newly diabetic NOD mice [48]. Furthermore, the GABA_A_ receptor agonist homotaurine inhibited effector T cell expansion at lower concentrations than GABA in early onset T1D in NOD mice but did not increase β-cell mass [49]. Homotaurine improved human islet β-cell replication and survival in human xenografts transplanted into STZ-treated NOD.*scid* mice [49]. Combining GABA with GAD65-alum restored euglycemia and improved the survival of transplanted neonatal pancreata to diabetic NOD mice [50]; however, the immunological microenvironment was not investigated. Our study is the first to examine the synergistic effects of GABA and GAD65-alum in children with T1D. Despite its short half-life and rapid metabolism, we observed long-term changes to the immune cytokine profile of our patients receiving oral GABA.

GAD65-alum has been investigated as a promising antigen-based immunotherapy for T1D in several clinical trials [51], but the cellular mechanism of action has not been fully elucidated. Recent reports from the DIAGNODE-1 study showed the efficacy of GAD65-alum delivery directly to the lymphatic organs to improve c-peptide levels [52]. β-cell preservation was partly mediated by increasing the pool of Tfh and CD8^+^ PD-1^+^ T cells [52]. Similar to our study, Claessens et al. performed an antigen recall assay with human PBMCs followed by HLA stratification [30]. In addition to GAD65, the antigens preproinsulin (PPI), islet antigen-2 (IA-2), and defective ribosomal product of the insulin gene (INS-DRIP) were used to stimulate PBMCs individually and in each possible combination. IA-2 and PPI resulted in the most robust T cell proliferation, but the stimulation indices for GAD65 and INS-DRIP were negative [30]. Thus, GABA treatment may be more effective in combination with multiple autoantigens and may explain in part why both GABA and GABA with GAD65-alum treatments inhibited Th1 responses compared to placebo but showed no significant difference between each other.

Another intriguing aspect of our study was the role of HLA alleles in response to GABA treatment. Recent reports show that GAD65-alum injections improved c-peptide retention best in HLA-DR3-DQ2, with the strongest response among those with HLA-DR3-DQ2 who lacked HLA-DR4-DQ8 [19]. This appears to contradict our results. However, this study was a post hoc analysis of 3 separate trials. One trial (*n* = 69) received the same interval and dosage of GAD65-alum as our study, but the other two trials (combined *n* = 452) received an additional booster at 90 days [15] or two boosters at 90 and 180 days [16]. The subsequent phase 2b DIAGNODE-2 trial, which confirmed in a pre-specified subgroup analysis efficacy in the HLA-DR3-DQ2 subpopulation, also differed from our study by using three monthly injections administered intralymphatically. Furthermore, we did not investigate the efficacy of GAD65-alum alone but only in combination with GABA. The GABA-induced anti-inflammatory biomarker response most pronounced in HLA-DR4-DQ8 patients demonstrated in our study, together with the previously demonstrated GAD65-alum response in HLA-DR3-DQ2 patients, might offer a complementary treatment approach to T1D covering both major HLA phenotypes.

## 5. Conclusions

In conclusion, we present evidence that GABA and GABA with GAD65-alum treatment may be efficacious in delaying Th1 effector responses in children with recent-onset T1D. Further optimization will be necessary for dosing, timing of disease onset, and duration of GABA treatment. Finally, we stress the importance of patient HLA haplotypes when assessing the efficacy of GABA and GAD65-alum treatment in future clinical trials. Future combinatorial treatments to enhance the efficacy of GABA and GAD65-alum treatment may involve the use of other immune modulators including Teplizumab, β-cell protecting therapies including Verapamil, and also dietary modulators that may affect immune responses [53,54,55].

## Figures and Tables

**Figure 1 biomedicines-11-01948-f001:**
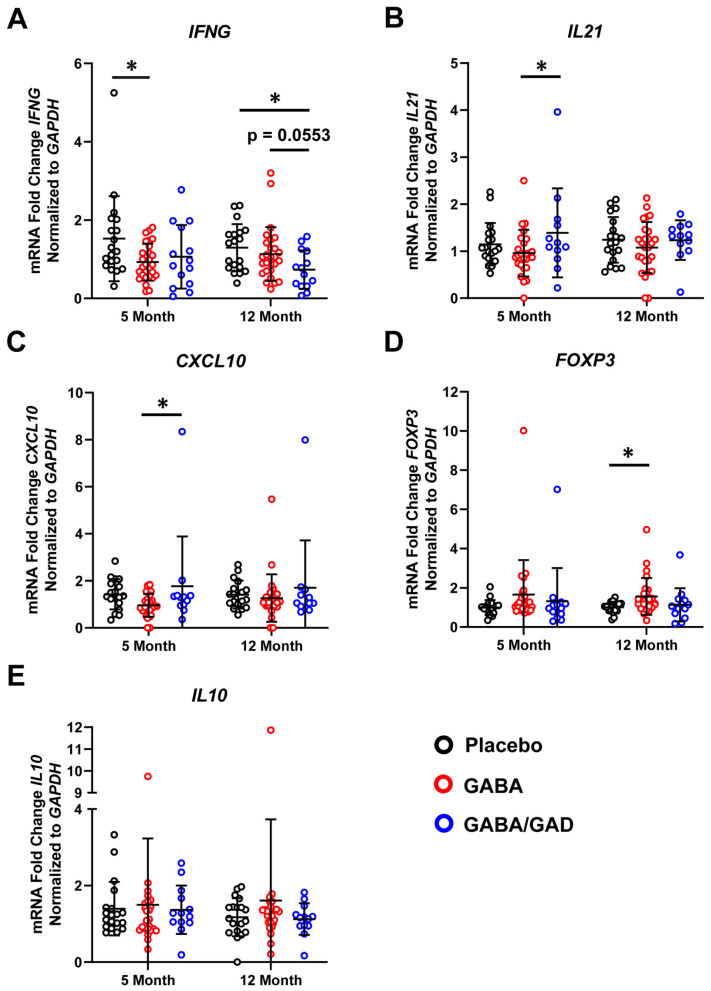
Treatment with GABA and GAD65-alum decreases Th1 cytokine responses of polyclonal-stimulated PBMCs. PBMCs from placebo, GABA, and GABA with GAD65-alum treatment groups were collected at 5- and 12 months and stimulated with Dynabeads for 72 h to examine changes in mRNA accumulation by polyclonal stimulation. TaqMan primers and probes specific for proinflammatory genes *IFNG* (**A**), *IL21* (**B**), and *CXCL10* (**C**) and anti-inflammatory genes *FOXP3* (**D**) and *IL10* (**E**) are shown. Placebo (*n* = 18), GABA (*n* = 24), GABA with GAD65-alum groups (*n* = 9). Data are shown as mean ± SD. Analyzed by Kruskal–Wallis tests between groups at each time point. * *p* < 0.05.

**Figure 2 biomedicines-11-01948-f002:**
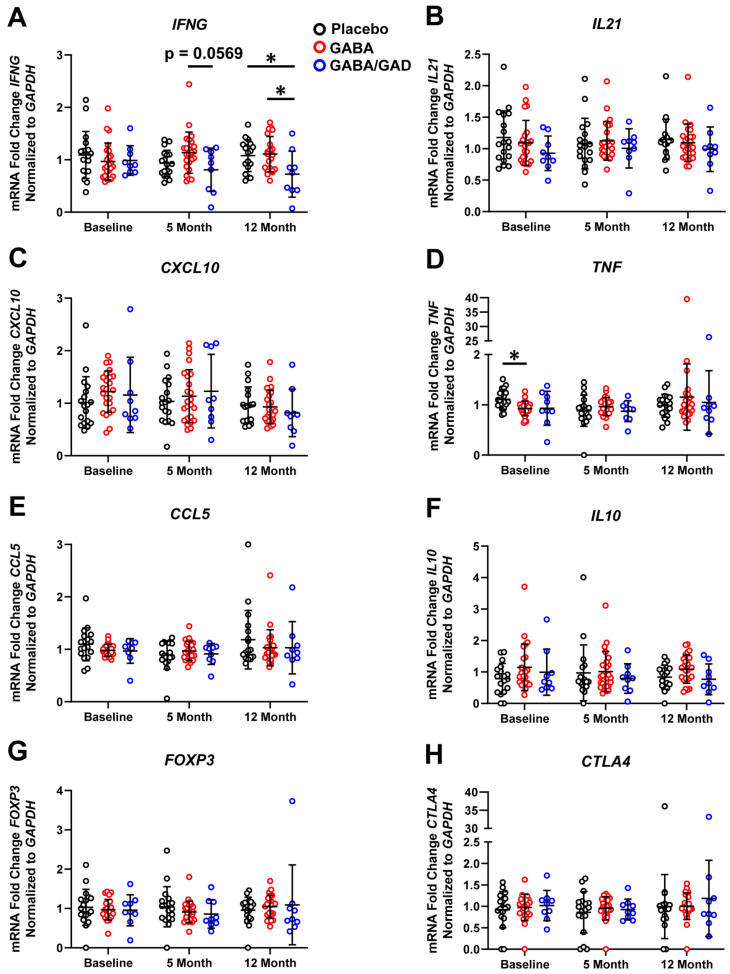
Treatment with GABA and GAD65-alum decreases GAD65-specific PBMC responses and *IFNG* mRNA accumulation. PBMCs from placebo, GABA, and GABA with GAD65-alum treatment groups were collected at baseline, 5 months, and 12 months and stimulated with rhGAD65 in an antigen-recall assay. mRNA was isolated 7 days after stimulation. Gene expression with TaqMan primers and probes specific for proinflammatory genes *IFNG* (**A**), *IL21* (**B**), *CXCL10* (**C**), *TNF* (**D**), and *CCL5* (**E**) and anti-inflammatory genes *IL10* (**F**), *FOXP3* (**G**) and *CTLA4* (**H**) are shown. Placebo group (*n* = 18), GABA group (*n* = 24), GABA with GAD65-alum group (*n* = 9). Data are shown as mean ± SD. Analyzed by Kruskal–Wallis tests between groups at each time point. * *p* < 0.05.

**Figure 3 biomedicines-11-01948-f003:**
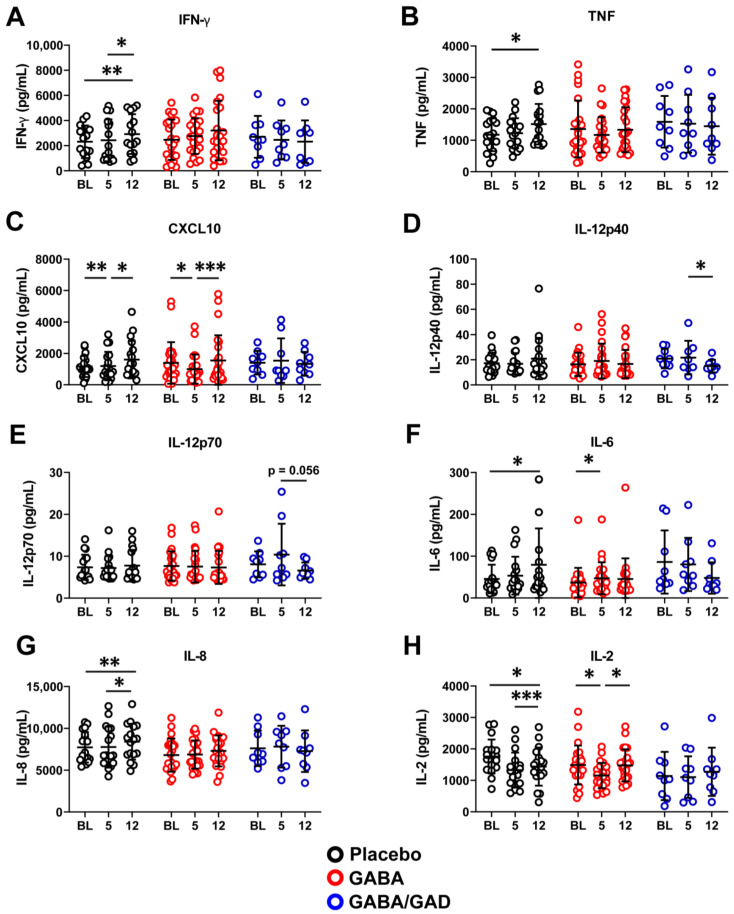
Treatment with GABA or a combination of GABA with GAD65-alum halts the progression of the Th1 effector phenotype. PBMCs from placebo, GABA, and GABA with GAD65-alum treatment groups were collected at baseline, 5 months, and 12 months and were stimulated with GAD65 for 7 days. Tissue culture supernatants were analyzed by Milliplex Luminex assay. Secreted IFN-γ (**A**), TNF (**B**), CXCL10 (**C**), IL-12p40 (**D**), IL-12p70 (**E**), IL-6 (**F**), IL-8 (**G**) and IL-2 (**H**) are shown. Placebo group (*n* = 18), GABA group (*n* = 24), GABA with GAD65-alum group (*n* = 9). Data are shown as mean ± SD. Analyzed by signed rank tests between time points. * *p* < 0.05, ** *p* < 0.01, *** *p* < 0.001.

**Figure 4 biomedicines-11-01948-f004:**
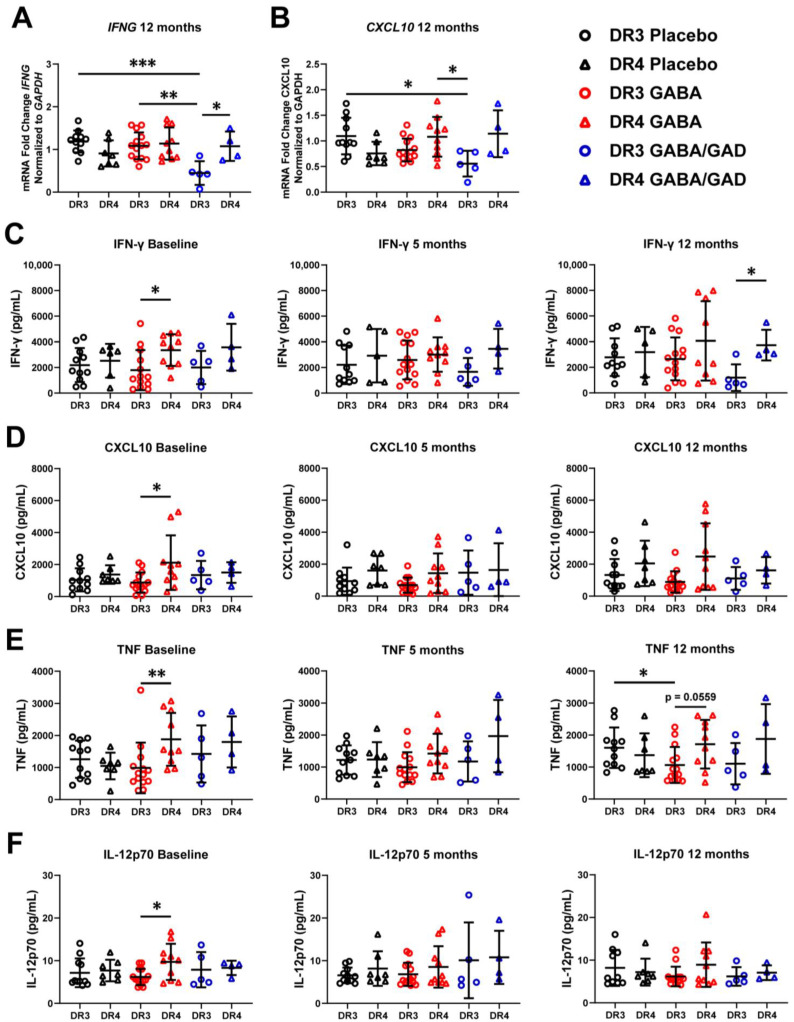
Patients with HLA-DR4/Other haplotype are predisposed to an inflammatory phenotype compared to patients with HLA-DR3-DQ2. Patient samples were genotyped and grouped according to HLA haplotype. Gene expression and cytokine/chemokine synthesis of HLA stratified PBMCs from placebo, GABA, and GABA with GAD65-alum treatment groups collected at baseline, 5 months, and 12 months following stimulation with GAD65 in an antigen-recall assay for 7 days. Gene expression of proinflammatory genes *IFNG* (**A**) and *CXCL10* (**B**). Supernatants were also collected 7 days post-stimulation and analyzed by Milliplex for IFN-γ (**C**), CXCL10 (**D**), TNF (**E**), and IL-12p70 (**F**) synthesis. Placebo group (HLA-DR3 N = *n*, HLA-DR4/Other *n* = 7), GABA group (HLA-DR3 *n* = 14, HLA-DR4/Other *n* = 10), GABA with GAD65-alum group (HLA-DR3 *n* = 5, HLA-DR4/Other *n* = 4). Data are shown as mean ± SD. Analyzed by Kruskal–Wallis tests between groups at each time point. * *p* < 0.05, ** *p* < 0.01, *** *p* < 0.0001.

**Table 1 biomedicines-11-01948-t001:** Demographic information of patients enrolled in this study.

Age (years)	Range (years)	4–18
	Mean ± SD (years)	11.4 ± 3.7
Sex	Male	*n* = 24 (47%)
	Female	*n* = 27 (53%)
Body mass index (BMI)	Range	14–28
	Mean ± SD	19.6 ± 3.6
Race	Caucasian	*n* = 47 (92%)
	African American	*n* = 3 (6%)
	Hispanic	*n* = 1 (2%)
Autoantibody positivity	Anti-GAD65*	*n* = 50 (98%)
	Anti-ICA 512	*n* = 34 (67%)
	Anti-Zinc Transporter 8	*n* = 48 (94%)
	Anti-Islet autoantibodies (IAA)	*n* = 23 (45)
Number of autoantibodies positive	1	*n* = 2 (4%)
	2	*n* = 2 (4%)
	3	*n* = 39 (76%)
	4	*n* = 8 (16%)

**Table 2 biomedicines-11-01948-t002:** TaqMan primers and probes used for mRNA expression analysis. All primer/probe sets were purchased from Applied Biosystems (Thermo Fisher).

Target	Item Number
*IFNG*	Hs00989291_m1
*IL21*	Hs00222327_m1
*CXCL10*	Hs01124251_g1
*FOXP3*	Hs01085834_m1
*IL10*	Hs00961622_m1
*TNF*	Hs01113624_g1
*BCL6*	Hs00153368_m1
*CCL5*	Hs00982282_m1
*CTLA4*	Hs03044418_m1
*GAPDH*	Hs03929097_g1

**Table 3 biomedicines-11-01948-t003:** Cytokine/chemokine panel analyzed by Milliplex.

EGF	IL-15	IL-7
Eotaxin	IL-17A	IL-8
G-CSF	IL-1Ra	CXCL10/IP-10
GM-CSF	IL-1α	CCL2/MCP-1
IFN-α2	IL-1β	CCL3/MIP-1α
IFN-γ	IL-2	CCL4/MIP-1β
IL-10	IL-3	CCL5/RANTES
IL-12p40	IL-4	TNFα
IL-12p70	IL-5	TNFβ
IL-13	IL-6	VEGF

**Table 4 biomedicines-11-01948-t004:** HLA genotypes of patients enrolled in this study.

HLA-DR3-DQ2		HLA-DR4/Other	
DR3-DR3, DQ2.5-DQ2.5	2	DR4-DR4, DQ8.1-DQ7.3	2
DR3-DR4, DQ2.5-DQ8.1	17	DR4-DR1, DQ8.1-DQ5.1	1
DR3-DR4, DQ2.5-DQ7.3	2	DR4-DR1, DQ8.1-DQ5.1	1
DR3-DR7, DQ2.5-DQ2.2	3	DR4-DR1, DQ8.1-DQ6.2	1
DR3-DR1, DQ2.5-DQ5.1	1	DR4-DR7, DQ8.1-DQ2.3	1
DR3-DR12, DQ2.5-DQ7.5	1	DR4-DR8, DQ8.1-DQ4.2	1
DR3-DR13, DQ2.5-DQ6.4	1	DR4-DR8, DQ8.1-DQ4.2	1
DR3-DR8, DQ2.5-DQ4.2	1	DR4-DR11, DQ8.1-DQ7.5	1
DR3-DR8, DQ2.5-DQ7.5	1	DR4-DR13, DQ8.1-DQ2.5	1
DR3-DR4, DQ2.5-Unknown	1	DR4-DR13, DQ8.1-DQ6.4	1
		DR4-DR13, DQ8.1-DQ6.4	1
		DR4-DR15, DQ8.1-DQ6.2	1
		DR4-DR4, DQ8-DQ7.3	1
		DR4-DR4, DQ7.3-DQ7.3	1
		DR4-DR13, DQ6.4-DQ7.2	1
		DR4-DR8, DQ4.2-DQ7.3	1
		DR7-DR1, DQ7.5-DQ2.2	1
		DR8-DR1, DQ4.2-DQ5.1	1
		DR9-DR9, DQ2.3-DQ2.3	1
		DR1-DR7, DQ5.1-DQ2.2	1
Total	30	Total	21

## Data Availability

The data that support the findings of this study are available upon reasonable request.

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
