# Peer review of "GABA and Combined GABA with GAD65-Alum Treatment Alters Th1 Cytokine Responses of PBMCs from Children with Recent-Onset Type 1 Diabetes"

_biomedicines, 2023, doi:10.3390/biomedicines11071948_

Round 1

Reviewer 1 Report

This is an interesting work with significant clinical implications on an important health problem.

The reviewer would like to offer a few points for consideration by the authors aiming at strengthening the paper:

1. Consider providing the inclusion and exclusion criteria for enrollment in the study.

2. Consider providing the rationale for sample size determination (ie: power analysis etc).

3. The reviewer would like to invite the authors to consider providing a short discussion about potentially dietary signaling agents that may be used towards immunomodulation. Along those lines a paper of potential interest (from a conceptual standpoint) that might be used is the following:

  1. Sikalidis AK (2015) Amino Acids and Immune Response: A role for cysteine, glutamine, phenylalanine, tryptophan and arginine in T-cell function and cancer? Pathol Oncol Res21(1):9-17. doi: 10.1007/s12253-014-9860-0.

Good job overall!

No issues with English language, proofreading is suggested.

Reviewer 2 Report

The authors submitted a research article in which they elucidated whether gamma-aminobutyric acid (GADA) and GAD65-alum reduce immune inflammatory responses in children with T1D. The authors found that GABA or GABA with GAD65-alum treatment can inhibit the synthesis of some proinflammatory cytokines and chemokines from polyclonal and rhGAD65-stimulated PBMCs. In addition to that, it turns out that patients with HLA-DR4/Other are predisposed to an elevated Th1 phenotype compared to HLA-DR3 patients, but respond well to GABA treatment. The authors suggested that HLA haplotypes can predict the efficacy of GABA and GAD65-alum treatment in patients with T1D. The fingings appear to be impressive and clinically important. However, I would like to make some comments to discuss.

1. The authors used descriptive statistics, while they declared that the aim of the study was "to investigate ... differences in response to GABA alone and GABA/GAD65-alum". It would be great to report not only differences in concentrations, but also factors contributing these differences and responces.

2. Ethical declaration should be more thoroughly reported including particularities of the population (IRB number, consent from children and their relatives, etc.).

3. Section Discussion. Please, give extensive explanation of clinical significance regarding cytokine responce difference in T1D patients.

Round 2

Reviewer 1 Report

The authors have reasonably addressed the reviewer's points.